# Point RCNN: An Angle-Free Framework for Rotated Object Detection

**Qiang Zhou** [1,†,*] **and Chaohui Yu** [2,†]

1    Alibaba Group, Hangzhou 311121, China
2    Alibaba Group, Beijing 100102, China; huakun.ych@alibaba-inc.com
\*    Correspondence: jianchong.zq@alibaba-inc.com; Tel.: +86-1307-369-6312
†    These authors contributed equally to this work.

**Abstract:** Rotated object detection in aerial images is still challenging due to arbitrary orientations, large scale and aspect ratio variations, and extreme density of objects. Existing state-of-the-art rotated object detection methods mainly rely on angle-based detectors. However, angle-based detectors can easily suffer from a long-standing boundary problem. To tackle this problem, we propose a purely angle-free framework for rotated object detection, called Point RCNN. Point RCNN is a two-stage detector including both PointRPN and PointReg which are angle-free. Given an input aerial image, first, the backbone-FPN extracts hierarchical features, then, the PointRPN module generates an accurate rotated region of interests (RRoIs) by converting the learned representative points of each rotated object using the MinAreaRect function of OpenCV. Motivated by RepPoints, we designed a coarse-to-fine process to regress and refine the representative points for more accurate RRoIs. Next, based on the learned RRoIs of PointRPN, the PointReg module learns to regress and refine the corner points of each RRoI to perform more accurate rotated object detection. Finally, the final rotated bounding box of each rotated object can be attained based on the learned four corner points. In addition, aerial images are often severely unbalanced in categories, and existing rotated object detection methods almost ignore this problem. To tackle the severely unbalanced dataset problem, we propose a balanced dataset strategy. We experimentally verified that re-sampling the images of the rare categories can stabilize the training procedure and further improve the detection performance. Specifically, the performance was improved from 80.37 mAP to 80.71 mAP in DOTA-v1.0. Without unnecessary elaboration, our Point RCNN method achieved new state-of-the-art detection performance on multiple large-scale aerial image datasets, including DOTA-v1.0, DOTA-v1.5, HRSC2016, and UCAS-AOD. Specifically, in DOTA-v1.0, our Point RCNN achieved better detection performance of 80.71 mAP. In DOTA-v1.5, Point RCNN achieved 79.31 mAP, which significantly improved the performance by 2.86 mAP (from ReDet's 76.45 to our 79.31). In HRSC2016 and UCAS-AOD, our Point RCNN achieved higher performance of 90.53 mAP and 90.04 mAP, respectively.

**Keywords:** rotated object detection; angle-based detector; angle-free framework; rotated region of interests (RRoIs); representative points





## 1. Introduction

Object detection has been a fundamental task in computer vision and has progressed dramatically in the past few years using deep learning. It aims to predict a set of bounding boxes and the corresponding categories in an image. Modern object detection methods of natural images can be categorized into two main categories: two-stage detectors, exemplified by Faster RCNN [1] and Mask RCNN [2], and one-stage detectors, such as YOLO [3], SSD [4], and RetinaNet [5].

Although object detection has achieved significant progress in natural images, it still remains challenging for rotated object detection in aerial images, due to the arbitrary

orientations, large scale and aspect ratio variations, and extreme density of objects [6]. Rotated object detection in aerial images aims to predict a set of oriented bounding boxes (OBBs) and the corresponding classes in an aerial image, which serves an important role in many applications, e.g., urban management, emergency rescue, precise agriculture, automatic monitoring, and geographic information system (GIS) updating [7,8]. Among these applications, antenna systems are very important for object detection, and many excellent examples [9–11] have been proposed.

Modern rotated object detectors can be divided into two categories in terms of the representation of OBB: angle-based detectors and angle-free detectors.

In angle-based detectors, an OBB of a rotated object is usually represented as a five-parameter vector $(x, y, w, h, \theta)$. Most existing state-of-the-art methods are angle-based detectors relying on two-stage RCNN frameworks [12–16]. Generally, these methods use an RPN to generate horizontal or rotated region of interests (RoIs), then a designed RoI pooling operator is used to extract features from these RoIs. Finally, an RCNN head is used to predict the OBB and the corresponding classes. Compared to two-stage detectors, one-stage angle-based detectors [17–21] directly regress the OBB and classify them based on dense anchors for efficiency. However, angle-based detectors usually introduce a long-standing boundary discontinuity problem [22,23] due to the periodicity of the angle and the exchange of edges. Moreover, the unit between $(x, y, w, h)$ and angle $\theta$ of the five-parameter representation is not consistent. These obstacles can cause the training to be unstable and limit the performance.

In contrast to angle-based detectors, angle-free detectors usually represent a rotated object as an eight-parameter OBB $(x_1, y_1, x_2, y_2, x_3, y_3, x_4, y_4)$, which denotes the four corner points of a rotated object. Modern angle-free detectors [24–27] directly perform quadrilateral regression, which is more straightforward than the angle-based representation. Unfortunately, although abandoning angle regression and the parameter unit is consistent, the performance of existing angle-free detectors is still relatively limited.

How to design a more straightforward and effective framework to alleviate the boundary discontinuity problem is the key to the success of rotated object detectors.

However, all the above methods use predefined (rotated) anchor boxes, whether angle-based or using angle-free methods. Compared to anchor boxes, representation points can provide more precise object localization, including shape and pose. Thus, the features extracted from the representative points may be less influenced by background content or uninformative foreground areas that contain little semantic information. In this paper, based on the learning of representative points, we propose a purely angle-free framework for rotated object detection in aerial images, called Point RCNN, which can alleviate the boundary discontinuity problem and attain state-of-the-art performance. Our Point RCNN is a two-stage detector and mainly consists of an RPN (PointRPN) and an RCNN head (PointReg), which are both angle-free. PointRPN serves as an RPN network. Given an input feature map, first, PointRPN learns a set of representative points for each feature point in a coarse-to-fine manner. Then, a rotated RoI (RRoI) is generated through the `MinAreaRect` function of OpenCV [28]. Finally, serving as an angle-free RCNN head, PointReg applies a rotate RoI Align [13,15] operator to extract RRoI features, and then refines and classifies the eight-parameter OBB of the corner points. In addition, the existing methods almost ignore the category imbalance in aerial images, and we propose to resample images of rare categories to stabilize convergence during training.

The main contributions of this paper are summarized as follows:

- We propose Point RCNN, a purely angle-free framework for rotated object detection in aerial images. Without introducing angle prediction, Point RCNN is able to address the boundary discontinuity problem.
- We propose PointRPN as an RPN network, which aims to learn a set of representative points for each object of interest, and can provide better detection recall for rotated objects in aerial images.

- We propose PointReg as an RCNN head, which can responsively regress and refine the four corners of the rotated proposals generated by PointRPN.
- Aerial images are usually long-tail distributed. We further propose to resample images of rare categories to stabilize training and improve the overall performance.
- Compared with state-of-the-art methods, extensive experiments demonstrate that our Point RCNN framework attains higher detection performance on multiple large-scale datasets and achieves new state-of-the-art performance.

## 2. Materials and Methods

### 2.1. Related Work

#### 2.1.1. Horizontal Object Detection

In the past decade, object detection has become an important computer vision task and has received considerable attention from academia and industry. Traditional methods use hand-crafted features (e.g., HoG, SIFT) to solve detection as classification on a set of candidate bounding boxes. With the development of deep convolutional neural networks (CNN), modern horizontal object detection methods can be mainly categorized into three types: two-stage detectors, one-stage detectors, and recent end-to-end detectors.

One line of research focuses on two-stage detectors [2,29–33], which first generate a sparse set of regions of interests (RoIs) with a region proposal network (RPN), and then perform classification and bounding box regression. While two-stage detectors still attract much attention, another line of research focuses on developing efficient one-stage detectors due to their much simpler and cleaner design [3–5,34–37], in which SSD [4] and YOLO [3] are the fundamental methods that use a set of pre-defined anchor boxes to predict object category and anchor box offsets. Note that anchors were first proposed in the RPN module of faster RCNN to generate proposals. Recently, more studies [38,39] that use bounding boxes for object detection have been reported. In addition, effort has been spent on designing anchor-free detectors [35,40]. FCOS [35] and Foveabox [40] use the center region of targets as positive samples. In addition, FCOS introduces the so-called centerness score to make non-maximum suppression (NMS) more accurate. The authors of [41] propose an adaptive training sample selection (ATSS) scheme to automatically define positive and negative training samples. PAA [42] involves a probabilistic anchor assignment strategy, leading to easier training compared to heuristic IoU hard-label assignment strategies. In addition to improve the assignment strategy of FCOS, efforts has been devoted to the detection features [43] and loss functions [44] to further boost anchor-free detector performance.

Very recently, several studies have proposed end-to-end frameworks for horizontal object detection by removing NMS from the pipeline. DETR [45] introduces a transformer-based attention mechanism to object detection. Essentially the sequence-to-sequence learning task in [46] was solved in parallel by a self-attention-based transformer rather than RNN. Deformable DETR [47] accelerates the training convergence of DETR by proposing to only attend to a small set of key sampling points. DeFCN [48] adopts a one-to-one matching strategy to enable end-to-end object detection based on a fully convolutional network with competitive performance. PSS [49] involves a compact and plug-in PSS head to eliminate heuristic NMS and achieve better performance.

#### 2.1.2. Rotated Object Detection

With the development of deep-learning technology, rotated object detection in aerial images has achieved great success in the past few years, especially with the release of the largest aerial image dataset DOTA [6], which has become a standard benchmark and has significantly boosted the development of rotated object detectors. In terms of the representation of the oriented bounding box (OBB), modern rotated object detectors can be mainly divided into two categories: angle-based detectors and angle-free detectors. As depicted in Figure 1, we show the main differences between angle-based detectors and angle-free detectors. Figure 1a shows the learning targets $(x, y, w, h, \theta)$ of angle-based detectors, where $(x, y)$ denote the coordinates of the center points, $(w, h)$ denote the shorter and longer edges

of the rotated bounding box, and $\theta$ denotes the angle between the longer edge and the horizontal axis. Figure 1b shows the learning targets $(x_1, y_1, x_2, y_2, x_3, y_3, x_4, y_4)$ of angle-free detectors, which represent the coordinates of four corner points of a rotated bounding box. Compared to angle-based detectors, angle-free detectors are more efficient since they are more straightforward and can alleviate the boundary discontinuity problem without introducing angle prediction.

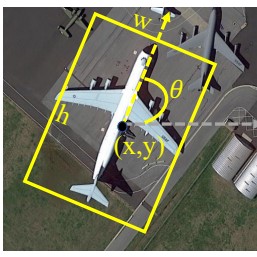
(a) Angle-based

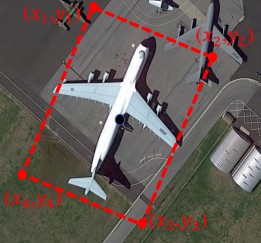
(b) Angle-free

**Figure 1.** Comparison of angle-based and angle-free detectors.

Angle-based detectors: Figure 2 illustrates three different methods for generating RRoIs: (a) and (b) denote two classical and mainstream RRoI generating methods. As shown in Figure 2a, one line of early research in the generation of RRoIs is the rotated region proposal network (rotated RPN) [17,18], which sets 54 anchors with different scales, angles, and aspect rations (three scales × six angles × three ratios) on each location to cover oriented objects. With the help of densely rotated anchors, the detection recall performance is thus improved. However, the introduction of massive rotated anchors increases the computational complexity and memory consumption, which limits the application of these methods. To tackle this issue, as shown in Figure 2b, the RoI transformer [13] proposes that RRoIs learn from horizontal RoIs by transforming default horizontal RoIs into RRoIs. The RoI transformer avoids introducing abundant anchors; however, it involves RPN, RoI alignment and regression, which are also complex processes. R$^2$CNN [12] proposes the detection of the horizontal and rotated bounding box simultaneously with multi-task learning. SCRDet [14] enhances features with an attention module and proposes an IoU-smooth $L_1$ loss to alleviate the loss discontinuity issue. SCRDet++ [23] extends SCRDet with image-level and instance-level de-noising modules to enhance the detection of small and cluttered objects. CSL [19] reformulates angle prediction from regression to classification to alleviate the discontinuous boundary problem. GWD [50] and KLD [51] propose a more efficient loss function for OBB regression. R3Det [20] proposes a refined single-stage rotation detector for fast and accurate object detection using a progressive regression approach from coarse to fine granularity. Constraint loss [52] proposes a decoupling modulation mechanism to overcome the problem of sudden changes in loss. S$^2$A-Net [21] proposes a single-shot alignment network to realize full feature alignment and alleviates the inconsistency between regression and classification. Recently, ReDet [15] has proposed the use of a rotation-equivariant network to encode rotation equivariance explicitly and presents a rotation-invariant RoI aligned to extract rotation-invariant features. The oriented RCNN [16] proposes a two-stage detector that consists of an oriented RPN for generating the RRoI and an oriented RCNN for refining the RRoI. Both ReDet and the oriented RCNN provide promising accuracy.

However, the boundary problem in the angle regression learning still causes training to be unstable and limits the performance. While angle-based detectors still find many applications, angle-free methods are receiving more and more attention from the research community.

Angle-free detectors: Textboxes++ [53] directly predict arbitrarily oriented word bounding boxes via a regression model by quadrilateral representation. ICN [24] proposes to directly estimate the four vertices of a quadrilateral to regress an oriented object based on an image pyramid and feature pyramid. RSDet [25] and gliding vertex [26] achieve more accurate rotated object detection via directly quadrilateral regression prediction. LR-

TSDet [8] proposes an effective tiny ship detector for low-resolution remote-sensing images based on horizontal bounding box regression. TPR-R2CNN [54] proposes an improved R2CNN based on a double-detection head structure and a three-point regression method. Recently, BBAVectors [27] have extended the horizontal keypoint-based object detector to an oriented object detection task. CFA [55] proposes a convex-hull feature adaptation approach for configuring convolutional features. Compared to angle-based methods, angle-free detectors are more straightforward and can alleviate the boundary problem to a large extent. However, the performance of current angle-free oriented object detectors is still relatively limited.

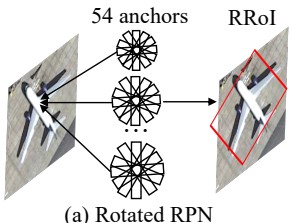
(a) Rotated RPN

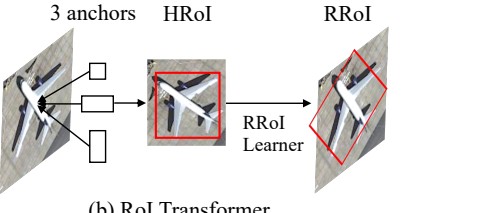
(b) RoI Transformer

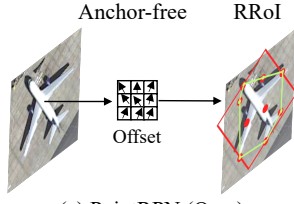
(c) PointRPN (Ours)

**Figure 2.** Comparison of different methods for generating rotated RoI (RRoI). (**a**) Rotated RPN places multiple rotated anchors with different angles, scales, and aspect ratios. (**b**) RoI transformer proposes an RRoI learner to model the RRoI from the horizontal RoI (HRoI) for each feature point based on 3 anchors. (**c**) Our proposed PointRPN generates accurate RRoI in an anchor-free and angle-free manner.

In this paper, we propose an effective angle-free framework for rotated object detection, called Point RCNN, which mainly consists of an RPN network (PointRPN) and an RCNN head (PointReg). Compared to the methods of Figure 2a,b, our proposed PointRPN generates accurate RRoIs in an anchor-free and angle-free manner. Specifically, PointRPN directly learns a set of implicit representative points for each rotated object. Based on these points, RRoIs can be easily attained with the `MinAreaRect` function of OpenCV. Without introducing anchors and angle regression, PointRPN becomes more efficient and accurate.

### 2.2. Methods

The overall structure of our Point RCNN is depicted in Figure 3. We start by revisiting the boundary discontinuity problem of angle-based detectors. Then, we describe the overall pipeline of Point RCNN. Finally, we elaborate the PointRPN and PointReg modules, and propose a balanced dataset strategy to rebalance the long-tailed datasets during training.

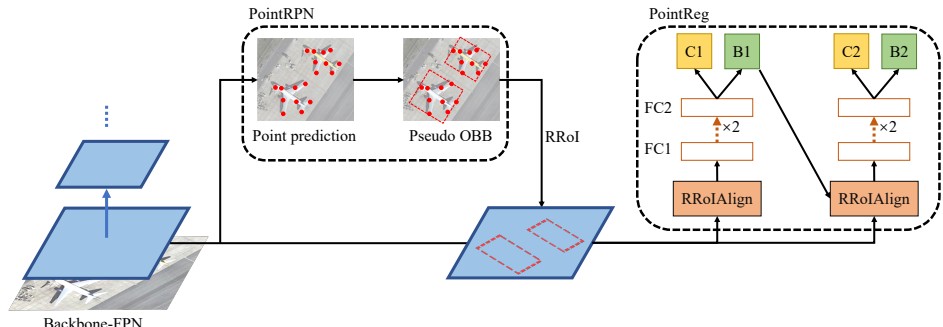

**Figure 3.** The overall pipeline of the proposed angle-free Point RCNN framework for rotated object detection. Point RCNN mainly consists of two modules: PointRPN for generating rotated proposals, and PointReg for refining for more accurate detection. "RRoI" denotes rotated RoI, "FC" denotes fully-connected layer, "C" and "B" represent the predicted category and rotated box coordinates of each RRoI, respectively.

### 2.2.1. Boundary Discontinuity Problem

The boundary discontinuity problem [22,23] is a long-standing problem that has existed in angle-based detectors. Taking the commonly used five-parameter OBB representation $(x, y, w, h, \theta)$ as an example, where $(x, y)$ represent the center coordinates, $(w, h)$ represent the shorter and longer edges of the bounding box, and $\theta$ represents the angle between the longer edge and the horizontal axis. As shown in Figure 4, when the target box is approximately square, a slight variation in edge length may cause $w$ and $h$ to swap, leading to a substantial variation in $\pi/2$ in angle $\theta$.

This boundary discontinuity issue in angle prediction will confuse the optimization of the network and limit the detection performance.

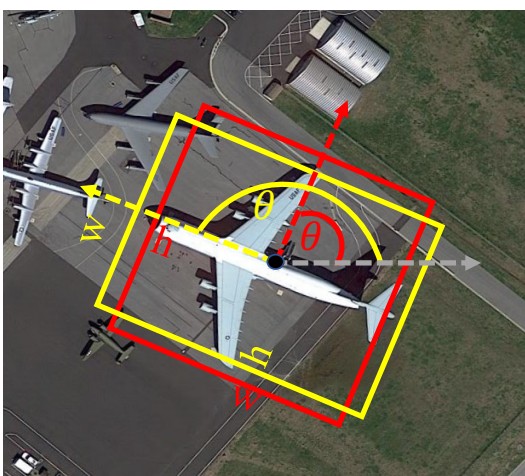

**Figure 4.** Boundary discontinuity problem of angle prediction. The red and yellow bounding boxes indicate two different targets. Although the two square-like targets have slightly different edge ($w$ and $h$) lengths, there is a huge gap between the angle target $\theta$.

### 2.2.2. Overview

To tackle the boundary problem in angle regression, in this paper, we propose a straightforward and efficient angle-free framework for rotated object detection. Instead of predicting the angle, as many previous angle-based two-stage methods do [13,15,16], our proposed Point RCNN reformulates the oriented bounding box (OBB) task as learning the representative points of the object in the RPN phase and modeling the corner points in the RCNN refine phase, which are both totally angle-free. Figure 5 shows the entire detection process, from the representative point learning to the final refined four corners of the oriented object.

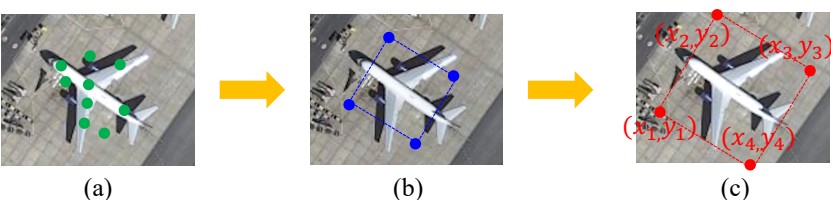

**Figure 5.** Illustration of the detection process of the Point RCNN framework. (**a**) denotes the predicted representative points with the PointRPN module. (**b**) denotes the conversion from the representative points to the rotated proposals. (**c**) denotes the refinement process of the corner points with the PointReg module.

The overall pipeline of Point RCNN is shown in Figure 3. During training, Backbone-FPN first extracts pyramid feature maps given an input image. Then, PointRPN performs representative points regression and generates a pseudo-OBB for the rotated RoI (RRoI). Finally, for each RRoI, PointReg regresses and refines the corner points and classifies them

for final detection results. Furthermore, we propose to resample images of rare categories to stabilize training and further improve the overall performance.

The overall training objective is described as:

$$\mathcal{L} = \mathcal{L}_{PointRPN} + \mathcal{L}_{PointReg}, \tag{1}$$

where $\mathcal{L}_{PointRPN}$ denotes the losses in PointRPN, and $\mathcal{L}_{PointReg}$ denotes the losses in PointReg. We will describe them in detail in the following sections.

### 2.2.3. PointRPN

Existing rotated object detection methods generate rotated proposals indirectly by transforming the outputs of RPN [1] and suffer from the boundary discontinuity problem caused by angle prediction. For example, Refs. [13,15] use an RoI transformer to convert horizontal proposals to rotated proposals with an additional angle prediction task. Unlike these methods, in this paper, we propose to directly predict the rotated proposals with representative point learning. The learning of points is more flexible, and the distribution of points can reflect the angle and size of the rotated object. The boundary discontinuity problem can thus be alleviated without angle regression.

Representative Points Prediction: Inspired by RepPoints [37] and CFA [55], we propose PointRPN to predict the representative points in the RPN stage. The predicted points can effectively represent the rotating box and can be easily converted to rotated proposals in subsequent RCNN stages.

As shown in Figure 6, PointRPN learns a set of representative points for each feature point. In order to make the features adapt more effectively to the representative points learning, we adopt a coarse-to-fine prediction approach. In this way, the features are refined with deformable convolutional networks (DCN) [56] and predicted offsets in the initial stage. For each feature point, the predicted representative points of the two stages are as follows:

$$
\begin{aligned}
\mathcal{R}^{init} &= \{(x_i^0 + \Delta x_i^0, y_i^0 + \Delta y_i^0)\}_{i=1}^K, \\
\mathcal{R}^{refine} &= \{(x_i^1 + \Delta x_i^1, y_i^1 + \Delta y_i^1)\}_{i=1}^K,
\end{aligned}
\tag{2}
$$

where $K$ denotes the number of predicted representative points and we set $K = 9$ by default. $\{(x_i^0, y_i^0)\}_{i=1}^K$ denotes the initial location, $\{(\Delta x_i^0, \Delta y_i^0)\}_{i=1}^K$ denote the learned offsets in the initial stage, and $\{(\Delta x_i^1, \Delta y_i^1)\}_{i=1}^K$ denote the learned offsets in the refine stage.

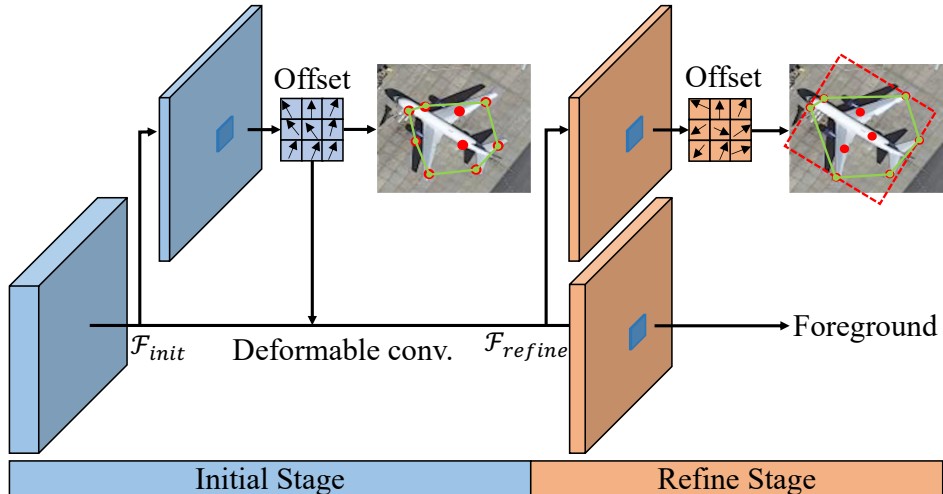

**Figure 6.** The structure of the proposed PointRPN. The red points are the learned representative points, and the green polygon represents the converted convex-hull. The red dotted OBB is converted from the representative points with the `MinAreaRect` function of OpenCV [28] for generating RRoI.

Label Assignment: PointRPN predicts representative points for each feature point in the initial and refine stages. This section will describe how we determine the positive samples among all feature points for these two stages.

For the initial stage (see the initial stage in Figure 6), we project each ground-truth box to the corresponding feature level $l_i$ according to its area, and then select the feature point closest to its center as the positive sample. The rule used for projecting the ground-truth box $b_i^*$ to the corresponding feature level is defined as:

$$l_i = \log_2 \left( \sqrt{\frac{w_i h_i}{s}} \right), \tag{3}$$

where $s$ is a hyper-parameter and is set to 16 by default. $w_i$ and $h_i$ are the width and height of the ground-truth box $b_i^*$. The calculated $l_i$ will be further limited to the range of $[3, 7]$, since we make predictions for the five feature levels of $(P_3, P_4, P_5, P_6, P_7)$. It is beneficial to optimize the overall detector by placing objects with different scales into different feature levels.

For the refine stage (see the refine stage in Figure 6), considering that the initial stage can already provide coarse prediction, we use the predicted representative points from the initial stage to help determine the positive samples for refined results. To be specific, for each feature point with its corresponding prediction $\mathcal{R}^{init}$, if the maximum convex-hull GIoU (defined in Equation (6)) between $\mathcal{R}^{init}$ and ground-truth boxes exceeds the threshold $\tau$, we select this feature point as a positive sample. We set $\tau = 0.1$ in all our experiments.

Optimization: The optimization of the proposed PointRPN is driven by classification loss and rotated object localization loss. The learning objective is formulated as follows:

$$\mathcal{L}_{PointRPN} = \lambda_1 {}^+\mathcal{L}_{loc}^{init} + \lambda_2 \, \mathcal{L}_{cls}^{refine} + \lambda_3 {}^+\mathcal{L}_{loc}^{refine}, \tag{4}$$

where $\lambda_1, \lambda_2$, and $\lambda_3$ are the trade-off parameters and are set to 0.5, 1.0, and 1.0 by default, respectively. ${}^+\mathcal{L}_{loc}^{init}$ denotes the localization loss of the initial stage. $\mathcal{L}_{cls}^{refine}$ and ${}^+\mathcal{L}_{loc}^{refine}$ denote the classification loss and localization loss of the refine stage. Note that the classification loss is only calculated in the refine stage, and the two localization losses are only calculated for the positive samples.

In the initial stage, the localization loss is calculated between the convex-hulls converted from the learned points $\mathcal{R}^{init}$ and the ground-truth OBBs, respectively. We use convex-hull GIoU loss [55] to calculate the localization loss:

$$ {}^+\mathcal{L}_{loc}^{init} = \frac{1}{N_{pos}^0} \sum_i \left( 1 - \text{CIoU} \left( \Gamma(\mathcal{R}_i^{init}), \Gamma(b_i^*) \right) \right), \tag{5}$$

where $N_{pos}^0$ indicates the number of positive samples of the initial stage. $b_i^*$ is the matched ground-truth OBB. CIoU represents the convex-hull GIoU between the two convex-hulls $\Gamma(\mathcal{R}_i^{init})$ and $\Gamma(b_i^*)$, which are differential and can be calculated as follows:

$$\text{CIoU}\left( \Gamma(\mathcal{R}_i^{init}), \Gamma(b_i^*) \right) = \frac{\left| \Gamma(\mathcal{R}_i^{init}) \cap \Gamma(b_i^*) \right|}{\left| \Gamma(\mathcal{R}_i^{init}) \cup \Gamma(b_i^*) \right|} - \frac{\left| \mathcal{P}_i \setminus \left( \Gamma(\mathcal{R}_i^{init}) \cup \Gamma(b_i^*) \right) \right|}{\mathcal{P}_i}, \tag{6}$$

where the first term denotes the convex-hull IoU, and $\mathcal{P}_i$ denotes the smallest enclosing convex object area of $\Gamma(\mathcal{R}_i^{init})$ and $\Gamma(b_i^*)$. $\Gamma(\cdot)$ denotes Jarvis's march algorithm [57] used to calculate the convex-hull from points.

The learning of the refine stage, which is responsible for outputting more accurate rotated proposals, is driven by both classification loss and localization loss. $\mathcal{L}_{cls}^{refine}$ is a standard focal loss [5], which can be calculated as:

$$\mathcal{L}_{cls}^{refine} = \frac{1}{N_{pos}^1} \sum_i \text{FL}(p_i, c_i^*), \tag{7}$$

$$\text{FL}(p_i, c_i^*) = \begin{cases} -\alpha(1 - p_i)^\gamma \log(p_i), & \text{if } c_i^* > 0; \\ -(1 - \alpha)p_i^\gamma \log(1 - p_i), & \text{otherwise,} \end{cases} \tag{8}$$

where $N_{pos}^1$ denotes the number of positive samples in the refine stage, $p_i$ and $c_i^*$ are the classification output and the assigned ground-truth category, respectively. $\alpha$ and $\gamma$ are hyper-parameters and are set to 0.25 and 2.0 by default. The localization loss $\mathcal{L}_{loc}^{refine}$ is similar to Equation (5) and can be formulated as:

$$^+\mathcal{L}_{loc}^{refine} = \frac{1}{N_{pos}^1} \sum_i \left(1 - \text{CIoU}\left(\Gamma(\mathcal{R}_i^{refine}), \Gamma(b_i^*)\right)\right). \tag{9}$$

With the refined representative points, the pseudo-OBB (see red-dotted OBB in Figure 6) is converted using the `MinAreaRect` function of OpenCV [28], which is then used for generating the RRoI for PointReg.

### 2.2.4. PointReg

Corner Points Refine: The rotated proposals generated by PointRPN already provide a reasonable estimate for the target rotated objects. To avoid the problems caused by angle regression and to further improve the detection performance, we refine the four corners of the rotated proposals in the RCNN stage. As shown in Figure 7, with the rotated proposals as input, we use an RRoI feature extractor [13,15] to extract the RRoI features. Then, given the RRoI features, two consecutive fully connected and ReLU layers are used to encode the RRoI features. Finally, two fully connected layers are responsible for predicting the class probability $P$ and refined corners $\mathcal{C}$ of the corresponding rotated object. The refined corner points can be represented as follows:

$$\mathcal{C} = \{(x_i + \Delta x_i, y_i + \Delta y_i)\}_{i=1}^4, \tag{10}$$

where $\{(x_i, y_i)\}_{i=1}^4$ denotes the four corner coordinates of the input rotated proposals, and we denote the corresponding four predicted corner offsets as $\{(\Delta x_i, \Delta y_i)\}_{i=1}^4$.

In PointReg, instead of directly performing angle prediction, we refine the four corners of the input rotated proposals. There are three advantages of adopting corner points refinement: (1) it can alleviate the boundary discontinuity problem caused by angle prediction; (2) the parameter units are consistent among the eight parameters $\{(x_i, y_i)\}_{i=1}^4$; and (3) it is possible to improve the localization accuracy using a coarse-to-fine approach.

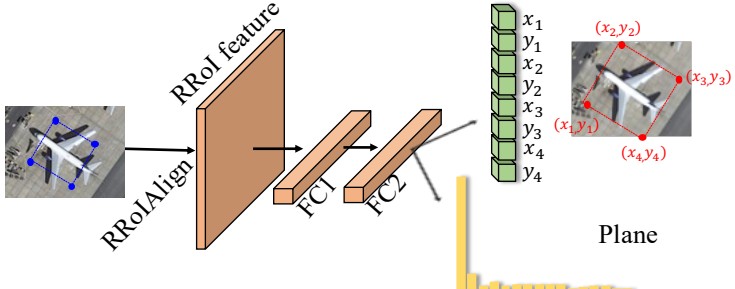

**Figure 7.** The diagram of the proposed PointReg. For simplicity, we only show the first stage of PointReg. The blue and red points represent the four corner points of the input RRoI and the refined results, respectively.

We can easily extend PointReg to a cascade structure for better performance. As shown in Figure 3, in the cascade structure, the refined rotated proposals of the previous stage are used as the input of the current stage.

Optimization: The learning of PointReg is driven by the classification loss and the rotated object localization loss:

$$\mathcal{L}_{PointReg} = \mu_1 \mathcal{L}_{cls} + \mu_2 {}^+\mathcal{L}_{loc}, \tag{11}$$

where $\mu_1$ and $\mu_2$ are the trade-off coefficients and are both set to 1.0 by default. $\mathcal{L}_{cls}$ indicates the classification loss, which is a standard cross-entropy loss:

$$\mathcal{L}_{cls} = -\frac{1}{N} \sum_i \sum_{c=0}^{C} Y_{i \to c} \log(P_i), \tag{12}$$

where $N$ denotes the number of training samples in PointReg, $C$ is the number of categories excluding the background, $P_i$ is the predicted classification probability of the $i_{th}$ RRoI. $Y_{i \to c} = 1$ if the ground-truth class of the $i_{th}$ RRoI is $c$; otherwise it is 0. ${}^+\mathcal{L}_{loc}$ represents the localization loss between the refined corners and the corners of the ground-truth OBB. We use $L_1$ loss to optimize the corner points refinement learning which can be calculated as:

$$ {}^+\mathcal{L}_{loc} = \frac{1}{N} \sum_i |\mathcal{C}_i - \vartheta(b_i^*)|, \tag{13}$$

where we let $\mathcal{C}_i (= \{(x_j, y_j)\}_{j=1}^4)$ denote the refined corners for the $i_{th}$ rotated proposal, let $b_i^* (= \{(x_j^*, y_j^*)\}_{j=1}^4)$ denote the corners of the matched ground-truth OBB. $\vartheta(b_i^*)$ denotes the permutation of four corners of $b_i^*$ with the smallest $L_1$ loss $|\mathcal{C}_i - \vartheta(b_i^*)|$, which can alleviate the sudden loss change issue in angle-free detectors. Note that ${}^+\mathcal{L}_{loc}$ is only calculated for positive training samples.

### 2.2.5. Balanced Dataset Strategy

The extremely non-uniform object densities of aerial images usually make the dataset long-tailed, which may cause the training process to be unstable and limit the detection performance. For instance, DOTA-v1.0 [6] contains 52, 516 ship instances but only 678 ground-track field instances [7]. To alleviate this issue, in this section, we propose a balanced dataset strategy. Specifically, we resample the images of rare categories, which was inspired by [58]. More concretely, first, for each category $c \in C$, we compute the fraction of images $F_c$ that contains this category. Then, we compute the category-level repeat factor for each category:

$$r_c = \max(1.0, \sqrt{\beta_{thr}/F_c}), \tag{14}$$

where $\beta_{thr}$ is a threshold which indicates that there will not be oversampling if "$F_c > \beta_{thr}$". Next, we compute the image-level repeat factor $r_I$ for each image $I$:

$$r_I = \max_{c \in C_I}(r_c), \tag{15}$$

where $C_I$ denotes the categories contained in image $I$. Finally, we can resample the images according to the image-level repeat factor. In other words, those images that contain long-tailed categories will have a greater chance of being resampled during training.

## 3. Results

In this section, we describe the dataset, evaluation protocol, implementation details, and demonstrate an overall evaluation and describe detailed ablation studies of the proposed method.

### 3.1. Datasets

To evaluate the effectiveness of our proposed Point RCNN framework, we performed experiments on four popular large-scale oriented object detection datasets: DOTA-v1.0 [6], DOTA-v1.5, HRSC2016 [59], and UCAS-AOD [60], which are widely used for rotated object detection. The statistic information comparison of these datasets is depicted in Table 1.

DOTA [6] is a large-scale and challenging aerial image dataset for oriented object detection with three released versions: DOTA-v1.0, DOTA-v1.5 and DOTA-v2.0. To compare

the performance with the state-of-the-art methods, we performed experiments on DOTA-v1.0 and DOTA-v1.5. DOTA-v1.0 contains 2806 images ranging in size from $800 \times 800$ to $4000 \times 4000$, and contains 188, 282 instances in 15 categories, abbreviated as: Bridge (BR), Harbor (HA), Ship (SH), Plane (PL), Helicopter (HC), Small vehicle (SV), Large vehicle (LV), Baseball diamond (BD), Ground track field (GTF), Tennis court (TC), Basketball court (BC), Soccer-ball field (SBF), Roundabout (RA), Swimming pool (SP), and Storage tank (ST). The dataset is divided into a training set, validation set, and test set, which account for one half, one sixth, and one third of the total dataset, respectively. DOTA-v1.5 is an updated version of DOTA-v1.0. It has the same images as DOTA-v1.0 but contains 402, 089 instances. DOTA-v1.5 has revised and updated the annotation of objects, where many small object instances about or below 10 pixels that were missed in DOTA-v1.0 have been additionally annotated. This is a more challenging dataset, which introduces a new category Container Crane (CC) and more small instances. For a fair comparison, we used the training set and validation set for training, and the test set was used to verify the performance of our model. The performances were obtained by submitting the prediction results to DOTA's evaluation server. The official evaluation protocol of the DOTA dataset in terms of the mAP was used.

**Table 1.** The statistic information comparison of the datasets. OBB denotes the oriented bounding box.

| Dataset | Source | Annotation | Categories | Instances | Images | Year |
|---------|--------|-----------|-----------|-----------|--------|------|
| UCAS-AOD [60] | Google Earth | OBB | 2 | 14,596 | 1510 | 2015 |
| HRSC2016 [59] | Google Earth | OBB | 1 | 2976 | 1061 | 2016 |
| DOTA-v1.0 [6] | multi source | OBB | 14 | 188,282 | 2806 | 2018 |
| DOTA-v1.5 | multi source | OBB | 15 | 402,089 | 2806 | 2019 |

HRSC2016 [59] is another popular dataset for oriented object detection. The images of this dataset were mainly collected from two scenarios, including ships on the sea and ships close to the shore. The dataset contains 1061 aerial images with size ranges from $300 \times 300$ to $1500 \times 900$, with most larger than $1000 \times 600$. There are more than 25 types of ships with large varieties in scale, position, rotation, shape, and appearance. This dataset can be divided into a training set, validation set and test set. There are 436 images, 181 images, and 444 images in the training set, validation set and test set, respectively. For a fair comparison, we used both the training and validation sets for training. The standard evaluation protocol of HRSC2016 dataset in terms of mAP was used.

UCAS-AOD [60] is another dataset for small oriented object detection with two categories (car and plane), which contains 1510 aerial images with 510 car images and 1000 airplane images. There are 14,596 instances in total, and the image size is approximately $659 \times 1280$. For a fair comparison, equivalent to the UCAS-AOD-benchmark (https://github.com/ming71/UCAS-AOD-benchmark, accessed on 29 March 2022), we also divided the dataset into 755 images for training, 302 images for validation, and 453 images for testing with a ratio of 5:2:3. The standard evaluation protocol of the UCAS-AOD dataset in terms of mAP was used.

### 3.2. Implementation Details

We implemented Point RCNN using the MMDetection tool-box [61]. We followed ReDet [15] to use ReResNet with RePFN as our backbone (ReR50-RePFN), which has shown the ability to extract rotation-equivariant features. We also verified with the more generalized transformer backbone (Swin-Tiny) to show the generalization and scalability of our Point RCNN framework.

For the DOTA dataset, following previous methods [13,15,21], we cropped the images to $1024 \times 1024$ with 824 pixels as a stride and we also resized the image to three scales $\{0, 5, 1.0, 1.5\}$ to prepare multi-scale data. Random horizontal flipping and random rotation ($[-45°, 45°]$) were adopted as the data augmentation for multi-scale training. For the HRSC2016 dataset, as in the previous method [15], we resized all the images to (800, 512), and we used random horizontal flipping as the data augmentation method during training.

For the UCAS-AOD dataset, following the UCAS-AOD-benchmark, we resized all the images to (800, 800) and only used the training set for training. We also used random horizontal flipping, HSV augment and random rotation as the data augmentation approach during training. Unless otherwise specified, we trained all the models with 19 epochs for DOTA, 36 epochs for HRSC2016, and 36 epochs for UCAS-AOD. Specifically, we trained all the models using the AdamW [62] optimizer with $\beta_1 = 0.9$ and $\beta_2 = 0.999$. The initial learning rate was set to 0.0002 with warming up for 500 iterations, with the learning rate decaying by a factor of 10 at each decay step. The weight decay was set to 0.05, and the mini-batch size was set to 16 (two images per GPU). We conducted the experiments on a server with 8 Tesla-V100 GPUs. The code will be released.

### 3.3. Main Results

We compared our Point RCNN framework with other state-of-the-art methods for four datasets: DOTA-v1.0, DOTA-v1.5, HRSC2016, and UCAS-AOD. As shown in Tables 2–5, without unnecessary elaboration, our Point RCNN demonstrated superior performance compared to state-of-the-art methods.

Results on DOTA-v1.0: As reported in Table 2, we first evaluated our method on the DOTA-v1.0 dataset and compared it with the popular and the state-of-the-art rotated object detection methods. We obtained the overall detection performance by submitting our results to the official DOTA-v1.0 evaluation server. In this comparison experiment, we compared many classic and impressive methods [13,19,21,23,26,27,55,63–65] and some state-of-the-art methods, e.g., Oriented RCNN [16] and ReDet [15].

As shown in Table 2, our Point RCNN method achieved new, state-of-the-art, detection performance against the comparison methods. More specifically, with the ReR50-ReFPN backbone, our Point RCNN improved the detection performance by 0.61 mAP against ReDet (form 80.10 to 80.71). Compared with Oriented RCNN, Point RCNN improved the performance by about 0.2 mAP. We observed that, with the proposed balanced dataset strategy, our Point RCNN was able to improve the performance by 0.34 mAP (from 80.37 to 80.71), which confirms that the extremely non-uniform rotated object densities of aerial images do limit detection performance.

In addition, we also evaluated our Point RCNN with the more generalized transformer backbone Swin-Tiny [66] (Swin-T). The Swin transformer [66] is a new vision transformer, which has been used as a general backbone of computer vision in recent years. Our proposed Point RCNN was able to further improve the performance by 0.61% (from 80.71 to 81.32), indicating that Point RCNN is scalable to general backbone networks.

Results on DOTA-v1.5: As reported in Table 3, we then evaluated our method on the DOTA-v1.5 dataset, which is a more challenging dataset, since it contains more categories and more small object instances. We obtained the overall detection performance by submitting our results to the official DOTA-v1.5 evaluation server. In this experiment, we compared some traditional strong two-stage oriented object detectors, e.g., Faster RCNN OBB (FR-O) [6], Mask RCNN [2], the Hybrid Task Cascade (HTC) [67] and state-of-the-art methods, including Oriented RCNN [16] and ReDet [15].

As shown in Table 3, our Point RCNN method achieved the new state-of-the-art detection performance on DOTA-v1.5 against the comparison methods. More specifically, our Point RCNN improved the detection performance by 2.51 mAP against ReDet (form 76.80 to 79.31), which represents a significant improvement for oriented object detection. Compared with Oriented RCNN, Point RCNN also significantly improved the performance by 2.86 mAP (from 76.45 to 79.31). We also observed that, with our proposed balanced dataset strategy, Point RCNN was able to further improve the performance by 0.57 mAP based on a high performance baseline (from 78.74 to 79.31). We also evaluated Point RCNN with the Swin-Tiny [66] (Swin-T) backbone. With the more generalized transformer backbone, our proposed Point RCNN was able to further improve the performance by **0.83** mAP (from 79.31 to 80.14), indicating that Point RCNN is scalable to general backbone networks and more challenging aerial image datasets.

**Table 2.** Performance comparisons on the DOTA-v1.0 test set (AP (%) for each category and overall mAP (%)). * denotes multi-scale training and testing, *† denotes the results of using our balanced dataset strategy. "R50" denotes ResNet-50, "R101" denotes ResNet-101, "R152" denotes ResNet-152, "H104" denotes Hourglass-104, "ReR50" denotes ReResNet-50, "Swin-T" denotes Swin Transformer Tiny.

| Method | Backbone | PL | BD | BR | GTF | SV | LV | SH | TC |
|---|---|---|---|---|---|---|---|---|---|
| RoI Trans. * [13] | R101-FPN | 88.64 | 78.52 | 43.44 | 75.92 | 68.81 | 73.68 | 83.59 | 90.74 |
| O$^2$-DNet * [63] | H104 | 89.30 | 83.30 | 50.10 | 72.10 | 71.10 | 75.60 | 78.70 | 90.90 |
| DRN * [64] | H104 | 89.71 | 82.34 | 47.22 | 64.10 | 76.22 | 74.43 | 85.84 | 90.57 |
| Gliding Vertex * [26] | R101-FPN | 89.64 | 85.00 | 52.26 | 77.34 | 73.01 | 73.14 | 86.82 | 90.74 |
| BBAVectors * [27] | R101 | 88.63 | 84.06 | 52.13 | 69.56 | 78.26 | 80.40 | 88.06 | 90.87 |
| CenterMap * [65] | R101-FPN | 89.83 | 84.41 | 54.60 | 70.25 | 77.66 | 78.32 | 87.19 | 90.66 |
| CSL * [19] | R152-FPN | 90.25 | 85.53 | 54.64 | 75.31 | 70.44 | 73.51 | 77.62 | 90.84 |
| SCRDet++ * [23] | R152-FPN | 88.68 | 85.22 | 54.70 | 73.71 | 71.92 | 84.14 | 79.39 | 90.82 |
| CFA * [55] | R-152 | 89.08 | 83.20 | 54.37 | 66.87 | 81.23 | 80.96 | 87.17 | 90.21 |
| S$^2$A-Net * [21] | R50-FPN | 88.89 | 83.60 | 57.74 | 81.95 | 79.94 | 83.19 | 89.11 | 90.78 |
| ReDet * [15] | ReR50-ReFPN | 88.81 | 82.48 | 60.83 | 80.82 | 78.34 | 86.06 | 88.31 | 90.87 |
| Oriented RCNN * [16] | R101-FPN | 90.26 | 84.74 | 62.01 | 80.42 | 79.04 | 85.07 | 88.52 | 90.85 |
| Point RCNN * (Ours) | ReR50-ReFPN | 82.99 | 85.73 | 61.16 | 79.98 | 77.82 | 85.90 | 88.94 | 90.89 |
| Point RCNN *† (Ours) | ReR50-ReFPN | 86.21 | 86.44 | 60.30 | 80.12 | 76.45 | 86.17 | 88.58 | 90.84 |
| Point RCNN *† (Ours) | Swin-T-FPN | 86.59 | 85.72 | 61.64 | 81.08 | 81.01 | 86.49 | 88.84 | 90.83 |

| | | BC | ST | SBF | RA | HA | SP | HC | mAP |
|---|---|---|---|---|---|---|---|---|---|
| RoI Trans. * [13] | R101-FPN | 77.27 | 81.46 | 58.39 | 53.54 | 62.83 | 58.93 | 47.67 | 69.56 |
| O$^2$-DNet * [63] | H104 | 79.90 | 82.90 | 60.20 | 60.00 | 64.60 | 68.90 | 65.70 | 72.80 |
| DRN * [64] | H104 | 86.18 | 84.89 | 57.65 | 61.93 | 69.30 | 69.63 | 58.48 | 73.23 |
| Gliding Vertex * [26] | R101-FPN | 79.02 | 86.81 | 59.55 | 70.91 | 72.94 | 70.86 | 57.32 | 75.02 |
| BBAVectors * [27] | R101 | 87.23 | 86.39 | 56.11 | 65.62 | 67.10 | 72.08 | 63.96 | 75.36 |
| CenterMap * [65] | R101-FPN | 84.89 | 85.27 | 56.46 | 69.23 | 74.13 | 71.56 | 66.06 | 76.03 |
| CSL * [19] | R152-FPN | 86.15 | 86.69 | 69.60 | 68.04 | 73.83 | 71.10 | 68.93 | 76.17 |

**Table 2.** *Cont.*

| Method | Backbone | PL | BD | BR | GTF | SV | LV | SH | TC |
|---|---|---|---|---|---|---|---|---|---|
| SCRDet++ * [23] | R152-FPN | 87.04 | 86.02 | 67.90 | 60.86 | 74.52 | 70.76 | 72.66 | 76.56 |
| CFA * [55] | R-152 | 84.32 | 86.09 | 52.34 | 69.94 | 75.52 | 80.76 | 67.96 | 76.67 |
| S$^2$A-Net * [21] | R50-FPN | 84.87 | 87.81 | 70.30 | 68.25 | 78.30 | 77.01 | 69.58 | 79.42 |
| ReDet * [15] | ReR50-ReFPN | 88.77 | 87.03 | 68.65 | 66.90 | 79.26 | 79.71 | 74.67 | 80.10 |
| Oriented RCNN * [16] | R101-FPN | 87.24 | 87.96 | 72.26 | 70.03 | 82.93 | 78.46 | 68.05 | 80.52 |
| Point RCNN * (Ours) | ReR50-ReFPN | 88.89 | 88.16 | 71.84 | 68.21 | 79.03 | 80.32 | 75.71 | 80.37 |
| Point RCNN *† (Ours) | ReR50-ReFPN | 88.58 | 88.44 | 73.03 | 70.10 | 79.26 | 79.02 | 77.15 | 80.71 |
| Point RCNN *† (Ours) | Swin-T-FPN | 87.22 | 88.23 | 68.85 | 71.48 | 82.09 | 83.60 | 76.08 | 81.32 |

**Table 3.** Performance comparisons on DOTA-v1.5 test set (AP (%) for each category and overall mAP (%)). * denotes multi-scale training and testing, *[†] denotes the results of using balanced dataset strategy. Note that the results of Faster RCNN OBB (FR-O) [6], RetinaNet OBB (RetinaNet-O) [5], Mask RCNN [2] and Hybrid Task Cascade (HTC) [67] are excerpted from ReDet [15]. The results of Oriented RCNN* and ReDet* with Swin-T-FPN backbone are our re-implementations based on their released official code. "R50" denotes ResNet-50, "R101" denotes ResNet-101, "ReR50" denotes ReResNet-50, "Swin-T" denotes Swin Transformer Tiny.

| Method | Backbone | PL | BD | BR | GTF | SV | LV | SH | TC | |
|---|---|---|---|---|---|---|---|---|---|---|
| RetinaNet-O [5] | R50-FPN | 71.43 | 77.64 | 42.12 | 64.65 | 44.53 | 56.79 | 73.31 | 90.84 | |
| FR-O [6] | R50-FPN | 71.89 | 74.47 | 44.45 | 59.87 | 51.28 | 68.98 | 79.37 | 90.78 | |
| Mask RCNN [2] | R50-FPN | 76.84 | 73.51 | 49.90 | 57.80 | 51.31 | 71.34 | 79.75 | 90.46 | |
| HTC [67] | R50-FPN | 77.80 | 73.67 | 51.40 | 63.99 | 51.54 | 73.31 | 80.31 | 90.48 | |
| OWSR * [68] | R101-FPN | - | - | - | - | - | - | - | - | |
| Oriented RCNN * [16] | R101-FPN | 87.20 | 84.67 | 60.13 | 80.79 | 67.51 | 81.63 | 89.74 | 90.88 | |
| ReDet * [15] | ReR50-ReFPN | 88.51 | 86.45 | 61.23 | 81.20 | 67.60 | 83.65 | 90.00 | 90.86 | |
| ReDet * [15] | Swin-T-FPN | 80.90 | 85.13 | 60.61 | 80.83 | 67.07 | 83.32 | 89.80 | 90.79 | |
| Point RCNN * (Ours) | ReR50-ReFPN | 83.40 | 86.59 | 60.76 | 80.25 | 79.92 | 83.37 | 90.04 | 90.86 | |
| Point RCNN *[†] (Ours) | ReR50-ReFPN | 83.12 | 86.55 | 60.84 | 82.43 | 80.60 | 83.39 | 90.01 | 90.88 | |
| Point RCNN * (Ours) | Swin-T-FPN | 83.88 | 85.22 | 60.76 | 79.40 | 81.64 | 83.48 | 89.98 | 90.75 | |
| Point RCNN *[†] (Ours) | Swin-T-FPN | 86.93 | 85.79 | 59.52 | 80.42 | 81.91 | 81.92 | 89.95 | 90.35 | |
| | | BC | ST | SBF | RA | HA | SP | HC | CC | mAP |
| RetinaNet-O [5] | R50-FPN | 76.02 | 59.96 | 46.95 | 69.24 | 59.65 | 64.52 | 48.06 | 0.83 | 59.16 |
| FR-O [6] | R50-FPN | 77.38 | 67.50 | 47.75 | 69.72 | 61.22 | 65.28 | 60.47 | 1.54 | 62.00 |
| Mask RCNN [2] | R50-FPN | 74.21 | 66.07 | 46.21 | 70.61 | 63.07 | 64.46 | 57.81 | 9.42 | 62.67 |
| HTC [67] | R50-FPN | 75.12 | 67.34 | 48.51 | 70.63 | 64.84 | 64.48 | 55.87 | 5.15 | 63.40 |
| OWSR * [68] | R101-FPN | - | - | - | - | - | - | - | - | 74.90 |
| Oriented RCNN * [16] | R101-FPN | 82.21 | 78.51 | 70.98 | 78.63 | 79.46 | 75.40 | 75.71 | 39.69 | 76.45 |
| ReDet * [15] | ReR50-ReFPN | 84.30 | 75.33 | 71.49 | 72.06 | 78.32 | 74.73 | 76.10 | 46.98 | 76.80 |
| ReDet * [15] | Swin-T-FPN | 86.04 | 78.69 | 75.35 | 77.38 | 78.48 | 75.41 | 79.51 | 61.95 | 78.20 |
| Point RCNN * (Ours) | ReR50-ReFPN | 87.45 | 84.50 | 72.79 | 77.32 | 78.29 | 77.48 | 78.92 | 47.97 | 78.74 |
| Point RCNN *[†] (Ours) | ReR50-ReFPN | 87.25 | 84.60 | 73.49 | 78.51 | 78.75 | 78.41 | 76.12 | 54.12 | 79.31 |
| Point RCNN * (Ours) | Swin-T-FPN | 87.00 | 84.65 | 70.70 | 77.87 | 78.32 | 79.50 | 74.35 | 63.80 | 79.46 |
| Point RCNN *[†] (Ours) | Swin-T-FPN | 85.72 | 85.84 | 68.57 | 76.35 | 78.79 | 81.24 | 78.64 | 69.23 | 80.14 |

Results on HRSC2016: We also verified our Point RCNN method on the HRSC2016 dataset, which contains many ship objects with arbitrary orientations. In this experiment, we compared our proposed Point RCNN method with some classic methods, e.g., RRPN [17], RoI-Trans. ref. [13], R$^3$Det [20], and S$^2$A-Net [21], and the state-of-the-art methods, Oriented RCNN [16] and ReDet [15]. Some methods were evaluated under the VOC2007 metric, while others were compared under the VOC2012 metric. To make a comprehensive comparison, we report the results for both metrics.

We report the experimental results in Table 4. We can observe that our Point RCNN method attained a new state-of-the-art performance under both the VOC2007 and VOC2012 metrics. Specifically, under the VOC2007 metric, our Point RCNN achieved 90.53 mAP, which exceeded the results for the comparison methods. It is worth noting that the Point RCNN significantly improved the performance by 0.90 and 0.93 mAP against ReDet and Oriented RCNN under the VOC2012 metric, respectively.

**Table 4.** Performance comparisons for the HRSC2016 test set. $mAP_{07}$ and $mAP_{12}$ indicate that the results were evaluated under VOC2007 and VOC2012 metrics (%), respectively. We report both results for fair comparison. "R50" denotes ResNet-50, "R101" denotes ResNet-101, "R152" denotes ResNet-152, "H34" denotes Hourglass-34, "ReR50" denotes ReResNet-50.

| Method | Backbone | $mAP_{07}$ (%) | $mAP_{12}$ (%) |
|---|---|---|---|
| RC2 [69] | VGG16 | 75.70 | - |
| RRPN [17] | R101 | 79.08 | 85.64 |
| $R^2$PN [18] | VGG16 | 79.60 | - |
| RRD [70] | VGG16 | 84.30 | - |
| RoI-Trans. [13] | R101-FPN | 86.20 | - |
| Gliding Vertex [26] | R101-FPN | 88.20 | - |
| $R^3$Det [20] | R101-FPN | 89.26 | - |
| DRN [64] | H34 | - | 92.7 |
| CenterMap [65] | R50-FPN | - | 92.8 |
| CSL [19] | R152-FPN | 89.62 | - |
| $S^2$A-Net [21] | R101-FPN | 90.17 | 95.01 |
| ReDet [15] | ReR50-ReFPN | 90.46 | 97.63 |
| Orient RCNN [16] | R101-FPN | 90.50 | 97.60 |
| Point RCNN (Ours) | ReR50-ReFPN | 90.53 | 98.53 |

Results on UCAS-AOD: The UCAS-AOD dataset consists of a large number of small rotated objects, which are often overwhelmed by complex scenes in aerial images. We evaluated our proposed Point RCNN method on UCAS-AOD and report the comparison results in Table 5. For a fair comparison, we report the results under the VOC2007 metric. As shown in Table 5, our proposed method achieved the best performance of **90.04** $mAP_{07}$, in which a value of 89.60 was obtained for the car detection, and a value of 90.48 was obtained for the airplane detection.

**Table 5.** Performance comparisons for the UCAS-AOD test set (AP (%) for each category and overall mAP (%)). All models were evaluated via the VOC2007 metric (%).

| Method | Backbone | Car | Airplane | $mAP_{07}$ (%) |
|---|---|---|---|---|
| R-Yolov3 [71] | Darknet53 | 74.63 | 89.52 | 82.08 |
| R-RetinaNet [5] | ResNet50 | 84.64 | 90.51 | 87.57 |
| RoI-Trans. [13] | ResNet50 | 88.02 | 90.02 | 89.02 |
| DAL [72] | ResNet50 | 89.25 | 90.49 | 89.87 |
| $S^2$A-Net [21] | ResNet50 | 89.56 | 90.42 | 89.99 |
| Point RCNN (Ours) | ReR50-ReFPN | 89.60 | 90.48 | 90.04 |

### *3.4. Ablation Study*

In this section, we report an ablation study of our proposed Point RCNN framework. If not specified, all the models were trained only on the training and validation set with a scale of 1.0 for simplicity, and were tested using multi-scale testing. The metric mAP was evaluated on the DOTA-v1.5 test set and obtained by submitting prediction results to DOTA-v1.5's evaluation server. In the following sections, we mainly elaborate the effectiveness of our angle-free Point RCNN framework, including PointRPN, PoingReg, and the balanced dataset strategy.

#### 3.4.1. Analysis of PointRPN

To analysis the efficiency of the proposed PointRPN, which serves as an RPN network, we evaluated the detection recall of PointRPN on the validation set of DOTA-v1.5. For simplicity, we trained the models on the training set with scale 1.0 and evaluated the detection recall on the validation set with scale 1.0 as well. The positive intersection over union (IoU) threshold was set to 0.5. We selected the top-300, top-1000, and top-

2000 proposals to calculate their recall values, respectively. The experimental results are reported in Table 6. We found that when the number of proposals reached 2000, as for the settings of many state-of-the-art methods [15,16], our PointRPN was able to attain 90.00% detection recall. When the number of proposals changed from top-2000 to top-1000, the detection recall value only dropped by 0.17%. Even if there were only top-300 proposals, our PointRPN was still able to achieve 85.93% detection recall. The high detection recall observed demonstrates that our angle-free PointRPN can alleviate the boundary discontinuity problem caused by angle prediction and effectively detect more oriented objects with arbitrary orientations in aerial images.

**Table 6.** Comparison of the detection recall results by varying the number of proposals of each image patch. The metric recall is evaluated on the DOTA-v1.5 validation set. $Recall_{300}$, $Recall_{1000}$, and $Recall_{2000}$ represent the detection recall of the top-300, top-1000, and top-2000 proposals, respectively.

| Method | $Recall_{300}$ (%) | $Recall_{1000}$ (%) | $Recall_{2000}$ (%) |
|---|---|---|---|
| PointRPN | 85.93 | 89.83 | 90.00 |

We also performed visualization analysis of PointRPN. As shown in Figure 8, we visualized some examples of the learned representative points of our PointRPN on the DOTA-v1.0 test set. The visualization results demonstrated that the proposed PointRPN was able to automatically learn the extreme points and the semantic key points of the rotated objects with arbitrary orientations, the large scale and aspect ratio variations, and the extreme non-uniform object densities.

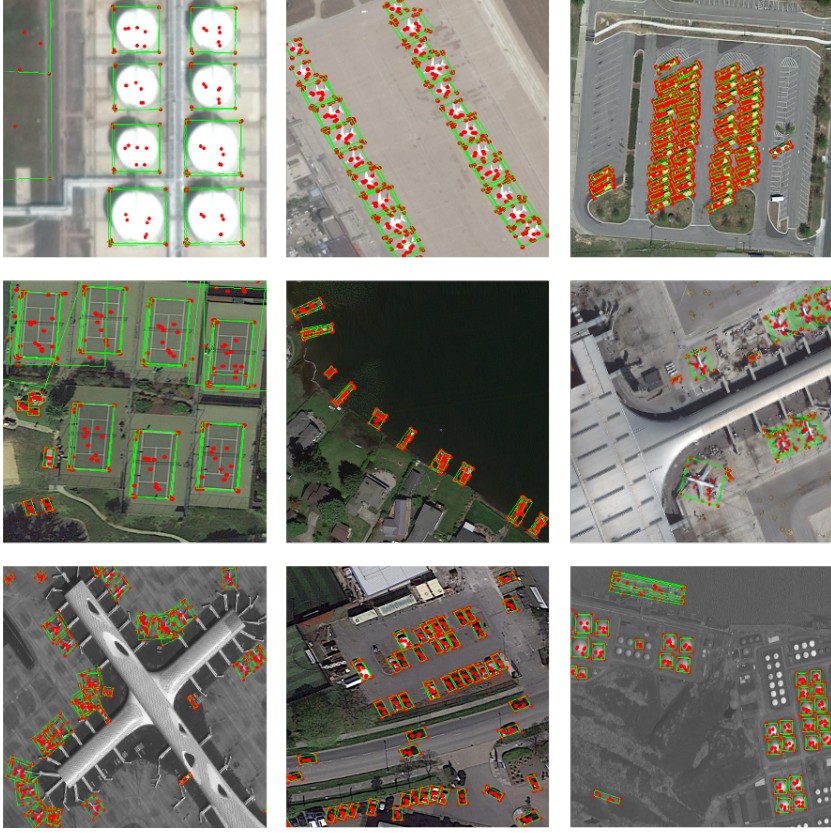

**Figure 8.** Visualization results of some examples of the learned representative points (red points) of PointRPN on the DOTA-v1.0 test set. The green oriented bounding boxes (OBBs) are the converted pseudo-OBBs via the `MinAreaRect` function of OpenCV. The score threshold was set to 0.001 without using NMS.

### 3.4.2. Effectiveness of PointReg

In this section, we provide an analysis of the effectiveness of the proposed PointReg module. We evaluated different OBB regression types of our PointReg and the results are reported in Table 7; compared to the five-parameter $(x, y, w, h, \theta)$ representation, the eight-parameter (also called corner points) $(x_1, y_1, x_2, y_2, x_3, y_3, x_4, y_4)$ regression type achieved higher detection performance (77.60 vs. 77.25) for oriented objects. In other words, our angle-free PointReg was shown to be capable of alleviating the boundary discontinuity problem caused by angle prediction and to achieve higher performance.

**Table 7.** Analysis of the effectiveness of OBB regression type of PointReg. The metric mAP was evaluated for the DOTA-v1.5 test set.

| Regression Type of PointReg | mAP (%) |
|:---:|:---:|
| $(x, y, w, h, \theta)$ | 77.25 |
| $(x_1, y_1, x_2, y_2, x_3, y_3, x_4, y_4)$ | 77.60 |

### 3.4.3. Analysis of Balanced Dataset Strategy

In this section, an analysis of the impact of the oversampling threshold $\beta_{thr}$ of the proposed balanced dataset strategy is provided. As shown in Table 8, we achieved the best detection accuracy of 77.60 mAP at $\beta_{thr} = 0.3$. Therefore, we set $\beta_{thr} = 0.3$ in all other experiments, unless otherwise stated.

**Table 8.** Comparison of detection accuracy by varying the oversampling threshold $\beta_{thr}$. The metric mAP was evaluated on the DOTA-v1.5 test set.

| Oversampling Threshold ($\beta_{thr}$) | mAP (%) |
|:---:|:---:|
| 0 | 73.52 |
| 0.1 | 76.49 |
| 0.2 | 77.44 |
| 0.3 | 77.60 |
| 0.4 | 77.48 |

### 3.4.4. Factor-by-Factor Experiment

To explore the effectiveness of each module of our proposed Point RCNN framework, we conducted a factor-by-factor experiment on the proposed PointRPN, PointReg and balanced dataset strategy. The results are depicted in Table 9. Each component had a positive effect, and all components were combined to obtain the best performance.

**Table 9.** Factor-by-factor ablation experiments. The detection performance was evaluated on the test set of DOTA-v1.5 dataset.

| Method | PointRPN | Balanced Dataset Strategy | PointReg | mAP (%) |
|:---:|:---:|:---:|:---:|:---:|
| Baseline | | | | 71.36 |
| | ✓ | | | 74.17 |
| Point RCNN | | ✓ | | 74.22 |
| | ✓ | ✓ | | 77.25 |
| | ✓ | ✓ | ✓ | 77.60 |

### 3.4.5. Visualization Analysis

We visualized some detection results for rotated objects for the DOTA-v1.0 test set. Figure 8 depicts some examples of the learned representative points of our PointRPN, which indicates that PointRPN was capable of learning the representative points of the rotated

object. Specifically, PointRPN was able to automatically learn the extreme points, e.g., the corner points of the rotated objects, and the semantic key points, e.g., the meaningful area of the rotated object.

Based on the reasonable prediction of high detection recall for the target rotated objects of PointRPN, our PointReg was able to continuously optimize and refine the corner points of the rotated objects. Some quantitative results for the DOTA-v1.0 test set are shown in Figure 9; the red points represent the corner points of the rotated objects learned by PointReg and the colored OBBs converted by the `MinAreaRect` function of OpenCV denote the final detection results. We also provide a visualization of the detection results for the UCAS-AOD and HRSC2016 datasets in Figures 10 and 11, respectively. The visualization results demonstrate the remarkable efficiency of our proposed angle-free Point RCNN framework for rotated object detection.

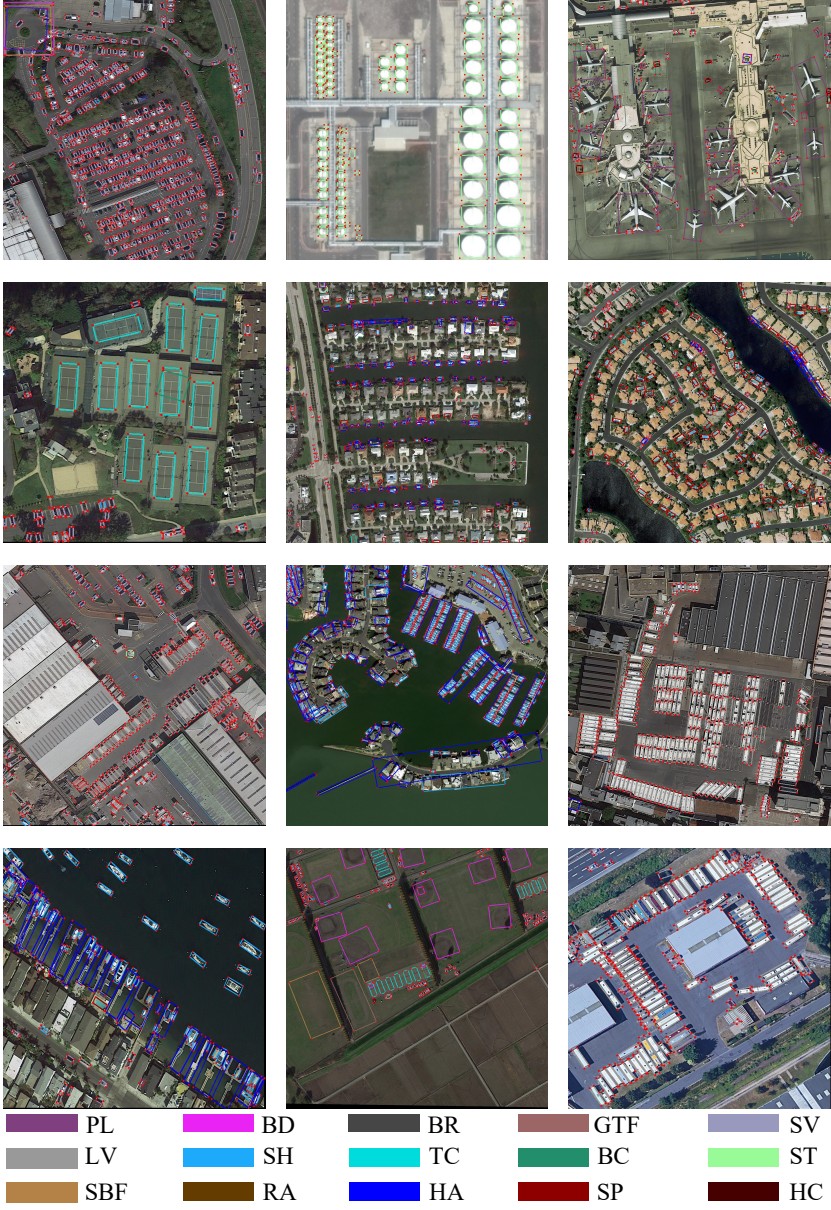

**Figure 9.** Visualization of the detection results of Point RCNN for the DOTA-v1.0 test set. The score threshold was set to 0.01. Each color represents a category. The red points and colored OBBs are the predicted corner points and the converted OBBs of PointReg.

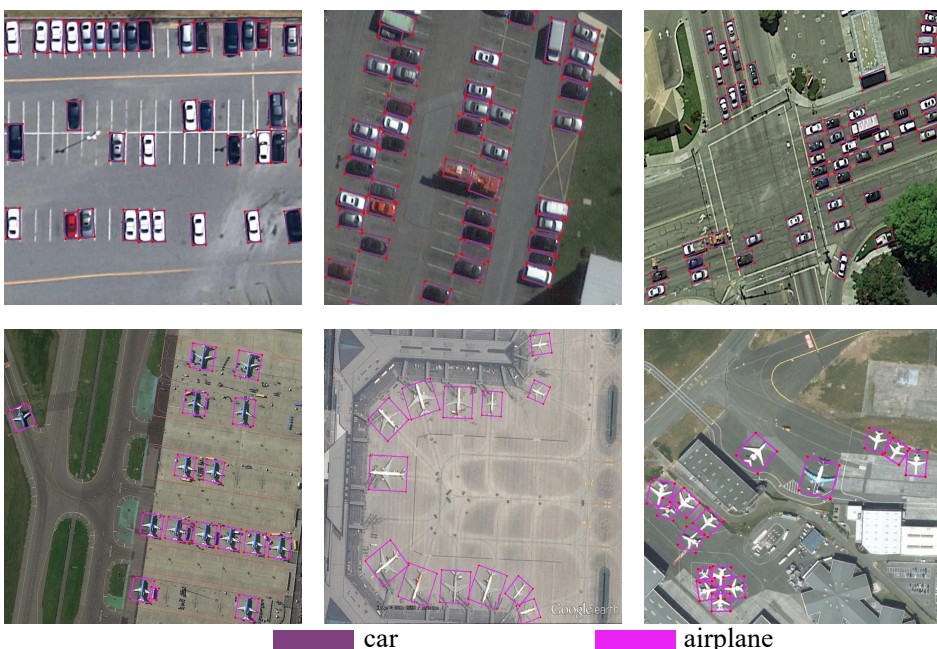

car          airplane

**Figure 10.** Visualization of the detection results of Point RCNN for the UCAS-AOD test set. The score threshold was set to 0.01. The red points and colored OBBs are the predicted corner points and the converted OBBs of PointReg.

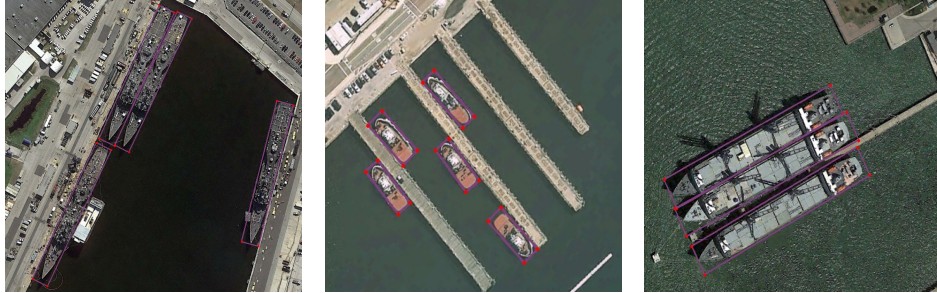

**Figure 11.** Visualization of the detection results of Point RCNN for the HRSC2016 test set. The score threshold was set to 0.01. The red points and colored OBBs are the predicted corner points and the converted OBBs of PointReg.

## 4. Discussion

Although the experiments undertaken substantiate the superiority of our proposed Point RCNN framework over state-of-the-art methods, our method did not perform well enough in some categories, e.g., PL (Plane) in the DOTA dataset, which requires further exploration. In addition, as with existing oriented object detectors, our Point RCNN also needs to use rotate non-maximum suppression (NMS) to remove duplicate results, which may mistakenly remove the true positive (TP) predictions and thus limit the final performance. Transformer-based methods [45] may provide potential solutions, which will be pursued in future work.

## 5. Conclusions

In this study, we revisited rotated object detection and proposed a purely angle-free framework for rotated object detection, named Point RCNN, which mainly consists of a PointRPN for generating accurate RRoIs, and a PointReg for refining corner points based on the generated RRoIs. In addition, we proposed a balanced dataset strategy to overcome the long-tailed distribution of different object classes in aerial images. Compared to existing rotated object detection methods, which mainly rely on angle prediction and

suffer from the boundary discontinuity problem, our proposed Point RCNN framework is purely angle-free and can alleviate the boundary problem without introducing angle prediction. Extensive experiments on multiple large-scale benchmarks demonstrated the significant superiority of our proposed Point RCNN framework against state-of-the-art methods. Specifically, Point RCNN achieved new state-of-the-art performances of 80.71, 79.31, 98.53, and 90.04 mAPs on DOTA-v1.0, DOTA-v1.5, HRSC2016, and UCAS-AOD datasets, respectively.

**Author Contributions:** Conceptualization, Q.Z. and C.Y.; methodology, Q.Z.; validation, Q.Z. and C.Y.; formal analysis, Q.Z.; investigation, Q.Z. and C.Y.; resources, C.Y.; data curation, C.Y.; writing—original draft preparation, Q.Z. and C.Y.; writing—review and editing, Q.Z. and C.Y.; visualization, C.Y.; supervision, Q.Z.; project administration, Q.Z.; funding acquisition, Q.Z. All authors have read and agreed to the published version of the manuscript.

**Funding:** This research received no external funding.

**Institutional Review Board Statement:** Not applicable.

**Informed Consent Statement:** Not applicable.

**Data Availability Statement:** Not applicable.

**Conflicts of Interest:** The authors declare no conflict of interest.

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
