# Peer review of "Point RCNN: An Angle-Free Framework for Rotated Object Detection"

_remotesensing, doi:10.3390/rs14112605_

Round 1

Reviewer 1 Report

Authors in this research work have introduced and investigated a rotated object detection and proposed a purely angle-free framework for rotated object detection, named Point RCNN, which mainly consists of a PointRPN for generating accurate RRoIs, and a PointReg for refining corner points based on the generated RRoIs. In addition, a balanced dataset strategy to overcome the long-tailed distribution of different object classes in aerial images has been described. The concept and idea of this work are interesting and the promising results have been introduced and experimentally validated. Although this work has been found attractive for remote sensing society, authors are requested to carefully address the following comments in the manuscript to improve its quality prior to final recommendation.

1) Authors in this research work have presented a purely angle-free framework for rotated object detection, called Point RCNN, this method should be discussed in more detail in the abstract section.

2) Abstract section is without any numerical achievements, please support this part by adding numerical findings.

3) In the abstract section authors have mentioned “we also experimentally verify that re-sampling the images of the rare categories will stabilize the training procedure and further improve the detection performance”. Please extend this sentence by providing more elaborations on it.

4) Antenna systems are very important for object detection. However, they have not been mentioned in the introduction section. So, please add this information in the introduction section to improve this part. Below are some helpful suggestions.

“Study on on-chip antenna design based on metamaterial-inspired and substrate-integrated waveguide properties for millimetre-wave and THz integrated-circuit applications”, Journal of Infrared, Millimeter, and Terahertz Waves 42 (1), 17-28, 2021.

"On-Chip Antenna Design Using the Concepts of Metamaterial and SIW Principles Applicable to Terahertz Integrated Circuits Operating over 0.6–0.622 THz" International Journal of Antennas and Propagation, Volume 2020, Article ID 6653095, 9 pages, https://doi.org/10.1155/2020/6653095.

“A new class of wideband microstrip falcate patch antennas with reconfigurable capability at circular‐polarization” Microw Opt Technol Lett. 2020; 1– 6, doi: 10.1002/mop.32529.

5) Fig.1 presents an interesting comparison of different methods for generating rotated RoI, so this comparison can be discussed in more depth to better understand readers.

6) Sections “2.2.1. Angle-based detectors” and “2.2.2. Angle-free detectors” can be supported with some figures, plots, charts to better understand readers.

7) The overall pipeline of the proposed angle-free Point RCNN framework for rotated object detection is exhibited in Fig.2, please mention how this framework has been derived and how it works?

8) Interesting equations have been presented in the manuscript, please describe how authors have extracted them? A proper explanation can be helpful for readers to follow the work.

9) Conclusion is short, authors can add some numerical results to this part. Also, the advantages of the proposed work can be highlighted here.

10) Reference part needs to be improved by a proper extension as per above mentioned suggestions.

11) Title is very short and seems very general, more information can be added in this part.

Reviewer 2 Report

The results are very promising since the reported perfomance is showing new state of the art achievement. Unfortunately I am not capable justify it since there is no open source project behind it. For this reason I sugest Authors focusing on e.g. github based open-project. Only in that situation other researchers can confirm these results. So I recommend to create an open source.

The main question addressed in the paper is the rotated object detection in aerial images. Authors claim that their method efficiently solves the problem of long-standing boundaries, therefore they proposed a purely angle-free framework for rotated object detection.

The application is original but methodology is rather state of the art. It does not fulfill any gap but it improves the performance of the existing methods. 

Reported better performance measured based on the state of the art. Methods and datasets.

The title of the paper is misleading since it tells us about “framework”. I am not capable to reconstruct the results, thus Authors should provide framework and test it with end users. Ones the are sure that other researchers can reproduce results the overall effort will be worth for publication.

Reviewer 3 Report

This paper presents a Point RCNN: An Angle-Free Framework for Rotated Object
Detection. The idea of using bounding feature points for training looks interesting even though the results are not very good. Comments to improve the manuscript are as follows;

  1. In the introduction authors have to further appeal the merits of point usage rather than bounding boxes.
  2.  Authors have to search for more works that use bounding boxes for learning and object detection. As the latest works; Low-Computational-Cost Algorithm for Inclination Correction of Independent Handwritten Digits on Microcontrollers, Intersection Detection Algorithm Based on Hybrid Bounding Box for Geological Modeling With Faults.
  3.  In Fig. 5, the difference between blue dots and red dots is not clear. Please, make the information more clear. On the other hand, in your paper the point detection process has to improved.
  4. In your experiments, the target objects have a feature of symmetry.  Does it work for non-symmetric object detection as well? Please add a discussion regarding this.

Round 2

Reviewer 1 Report

Authors have successfully addressed the reviewers' comments. So, the quality of the manuscript has been improved as per requested. No more technical comments are available. 

Reviewer 2 Report

Paper in current form is ready for publication.

Reviewer 3 Report

Appreciate the revisions by authors. This paper is now at the acceptable level.